# LLM-ORBench: Designing a Benchmark Dataset for Complex Ontology-Based Reasoning Tasks in Large Language Models

## Abstract

Large Language Models (LLMs) are increasingly applied to tasks requiring complex reasoning, yet their capabilities in formal logical reasoning remain underexplored. Existing benchmarks often focus on pattern recognition and fail to adequately assess symbolic reasoning, abstraction, or noise handling. To address this, we introduce *LLM-ORBench*, a benchmark framework for evaluating LLMs on structured, ontology-based tasks with verifiable multi-step inferences generated by a symbolic reasoner. The framework combines natural language and formal SPARQL questions, and systematically removes domain knowledge (i.e., abstraction) to isolate formal logical reasoning. We evaluated *GPT-5-mini*, *DeepSeek-V3-0324*, and *LLaMA-4-Maverick-17B-128E-Instruct* on two ontologies—*Family* and *OWL2Bench*—across binary and open-ended question-answering tasks. Our results show that reasoning complexity, abstraction, and question type strongly affect accuracy and reliability, with reasoning on abstracted tasks producing low accuracy and overconfidence, and open-ended tasks exhibiting substantial hallucination rates.

## 1 Introduction

Large Language Models (LLMs) (Brown et al., 2020) have demonstrated impressive capabilities in natural language tasks, excelling at understanding and generating human-like text, which makes them effective for applications such as chatbots and language translation (Kau et al., 2024). However, they also have notable limitations, including hallucinations (i.e., outputs that are factually incorrect or logically inconsistent (Chen et al., 2025)), as well as a lack of domain-specific knowledge and poor interpretability, which can diminish trust in their outputs (Pan et al., 2024). Hallucinations raise a critical question: are LLMs truly capable of reasoning, or are they merely producing plausible-sounding text? While some benchmarks indicate strong performance on reasoning tasks, others highlight significant failures, particularly in commonsense reasoning and planning scenarios (Valmeekam et al., 2022). Moreover, many existing benchmarks are still overly simplistic and do not adequately address complex reasoning challenges. To ensure that LLMs are reliable and trustworthy, especially for knowledge-intensive applications, more rigorous evaluation methods are essential (Chang et al., 2024).

Symbolic AI approaches, particularly ontologies (see Section 3.1.1 for the definition), emphasize logical reasoning and explicit knowledge representation, excelling in tasks that require precise inference and long-term memory (Hammond & Leake, 2023). Building on these strengths, the emerging field of neurosymbolic AI (Garcez & Lamb, 2019; Besold et al., 2017) seeks to combine the pattern-recognition capabilities of neural networks with the structured reasoning of symbolic logic. Within this context, LLMs are increasingly explored as potential components of reasoning systems, though their ability to perform rigorous logical inference over formal ontologies remains largely underexplored.

Standard ontology reasoners provide sound and complete inference within well-defined logical frameworks, but they are limited in handling noisy, incomplete, or ambiguous knowledge, and they lack integration with natural language. LLMs, by contrast, can interface seamlessly with human language, approximate reasoning beyond strict logical entailments, and incorporate broad world

knowledge acquired during pretraining. These properties make them a promising complement to traditional reasoners, enabling more flexible, accessible, and knowledge-rich forms of ontology reasoning.

Existing benchmarks, such as ProofWriter (Tafjord et al., 2020), ProntoQA (Saparov & He, 2023), and LogicBench (Parmar et al., 2024), have advanced our understanding of LLM reasoning capabilities (McIntosh et al., 2025; Fodor, 2025). However, they exhibit several limitations. ProofWriter is confined to synthetic rules and templated English, restricting evaluation to small-scale, shallow, and noise-free scenarios. ProntoQA advances formal proof–grounded evaluation but remains limited to synthetic ontologies, short proofs, and deterministic yes/no answers, offering little insight into richer reasoning behaviors. LogicBench provides fine-grained rule-level testing but only for single-rule reasoning in small-scale settings, lacking multi-step or large-scale ontology evaluation.

To address these gaps, we present *LLM-ORBench*, a novel benchmark framework for systematically evaluating LLMs on complex ontology-based reasoning tasks. The questions and answers in our benchmark are derived from verifiable multi-step inferences generated by a symbolic reasoner. Our study investigates whether LLMs can consistently and accurately perform multi-step reasoning over ontologies and how effectively they manage abstraction. We evaluate their ability to produce correct answers through logical inference rather than relying on domain-specific knowledge. The main contributions of this work are:

1. *LLM-ORBench* — a benchmark framework specifically designed for evaluating LLMs on complex ontology-based reasoning tasks.

2. A systematic methodology for introducing complexity through abstraction to isolate formal reasoning from domain knowledge cues.

3. An empirical comparison of multiple LLMs and a well-known symbolic reasoner across diverse ontology reasoning tasks and benchmark variations.

4. Novel insights into the generalization, robustness, and limitations of LLMs in formal logic.

## 2 RELATED WORK

Evaluating the reasoning capabilities of large language models (LLMs) has become an important area of research. While numerous benchmarks have been introduced, many remain narrow in scope and fail to systematically assess multi-step reasoning or the ability to address open-ended questions in formal logic. In this section, we review representative datasets and frameworks developed to quantify LLM reasoning performance, highlighting their strengths and limitations and motivating the design of our proposed benchmark.

Tafjord et al. (2020) proposed *ProofWriter*, a generative model that produces both the implications of a theory and natural language proofs supporting them. Given facts, rules, and a question in natural language, ProofWriter answers the question and generates a proof. They evaluated three tasks: (i) proof generation, (ii) implication enumeration, and (iii) abduction (single fact) over natural language theories. Experiments used the RuleTaker D* datasets (Clark et al., 2020) and two new variants: CWA (Closed-World Assumption), which corrects minor issues with negation, and OWA (Open-World Assumption), which adopts an open-world assumption. Each dataset contains a theory, a question, an answer (True/False/Unknown), and all possible proofs. The datasets comprise five subsets ($D0$–$D5$), each with 100,000 questions. Theories and questions are expressed in templated English. Questions may be positive or negated facts, and answers are balanced across True/False (and Unknown in OWA). Reasoning depth ranges from $D = 0, 1, 2, 3, 5$. ProofWriter achieves up to $+9\%$ higher proof accuracy than prior methods, and generalizes to deeper proofs and out-of-domain problems. However, since the proofs are generated using synthetic rules (EntailmentBank), the reasoning is restricted by predefined templates. Consequently, ProofWriter is confined to synthetic, small-scale scenarios with controlled English ontologies, shallow proof depths, and lacks evaluation under noisy conditions.

Saparov & He (2023) introduced *Proof and Ontology-Generated Question-Answering* (*PRONTOQA*), a synthetic dataset designed to systematically evaluate LLMs' reasoning capabilities beyond simple label prediction. Each example in *PRONTOQA* is generated from a controlled ontology and is associated with a unique formal proof, which is then translated into a natural language Chain-of-

Thought (CoT) example containing a context, a query, the reasoning steps, and the final label. While *PRONTOQA* represents a valuable advance in the assessment of logical reasoning in LLMs, it is still limited to synthetic, small-scale ontologies and relatively short proofs within tightly controlled natural language. Furthermore, the deterministic yes/no answers leave little opportunity to assess free-form or nuanced reasoning. The complexity of real-world applications demands benchmarks that assess neurosymbolic AI models across more challenging dimensions, including realistic and large-scale ontologies, higher branching factors, multi-step reasoning, robustness under ontology or symbol variation, and richer quantitative and qualitative evaluation of reasoning behavior.

Parmar et al. (2024) present *LogicBench*, a natural language question-answering dataset that evaluates the application of 25 distinct inference rules. The benchmark enables fine-grained, rule-specific evaluation of LLMs—including GPT-4, ChatGPT, Gemini, LLaMA-2, and Mistral under chain-of-thought prompting. Experimental results show that even state-of-the-art models struggle with certain inference patterns, particularly those involving complex reasoning and negation. However, LogicBench is similarly limited to synthetic, small-scale ontologies and focuses exclusively on single-rule reasoning, lacking evaluation of extended multi-step reasoning that combines multiple inference rules. Moreover, while the benchmark reports detailed quantitative results, it provides limited qualitative analysis.

To sum up, LogicBench, ProntoQA, and ProofWriter each test aspects of logical or deductive reasoning in LLMs, but they remain limited by synthetic design, shallow or template-based reasoning chains, and narrow linguistic or domain coverage. As a result, they fall short of capturing the complexity and diversity of real-world reasoning. Our work addresses this gap by systematically generating questions that require complex, multi-step reasoning through combinations of inference rules and abstraction. Table 1 presents a comparative overview of LLM-ORBench and selected reasoning benchmarks based on the following features:

1. **Complex Reasoning Chains**: Multi-step reasoning that involves multiple inference steps, with dependencies between steps.

2. **Robustness to Abstracted Data**: Testing robustness to abstracted inputs by systematically removing domain knowledge to isolate formal logical reasoning.

3. **Quantitative & Qualitative Metrics**: Inclusion of both objective measures (i.e., accuracy) and qualitative analysis (i.e., confidence, hallucination).

4. **Open-ended Questions**: Ability to handle questions requiring free-form responses rather than binary or multiple-choice answers. Open-ended questions better evaluate LLM reasoning because they require free-form responses, reducing the chance of correct answers by random selection. Binary or multiple-choice formats may mask hallucinations, whereas open-ended prompts reveal the model's true reasoning capabilities.

## 3 LLM-ORBENCH FRAMEWORK

### 3.1 LOGIC TYPES

#### 3.1.1 DESCRIPTION LOGIC

An ontology is a formal, explicit specification of a shared conceptualization of a domain. It defines concepts, categories, relationships, and rules that describe and structure knowledge within that domain, enabling common understanding and interoperability between systems and people (Gruber, 1993).

The ontologies used throughout this paper are developed using `OWL DL`, which is based on *description logic* (Krötzsch et al., 2013), a decidable fragment of *first-order logic*. Description logic distinguishes between the *TBox*, which defines classes and relationships between concepts, and the *ABox*, which contains individuals and their property or membership assertions.

This logic supports the definition of classes, individuals, and properties, enabling precise modeling of domain knowledge. For example, a membership assertion may state that `John` is a `Student`, while a property assertion may state that `John hasAdvisor ProfSmith`. `OWL DL` allows for automated reasoning, including consistency checking, inference of class hierarchies, and instance

| Benchmark | Feature | Support | Note |
|---|---|:---:|---|
| ProofWriter | Complex Reasoning Chains | ✓ | proofs up to 5 steps |
| | Robustness to Abstracted Data | ✗ | clean inputs only |
| | Quantitative / Qualitative | ★ | quantitative only |
| | Open-ended Questions | ✗ | true/false/unknown |
| ProntoQA | Complex Reasoning Chains | ✓ | sequence of 1-step *modus ponens* |
| | Robustness to Abstracted Data | ✓ | supports abstractions |
| | Quantitative / Qualitative | ★ | quantitative only |
| | Open-ended Questions | ✗ | true/false |
| LogicBench | Complex Reasoning Chains | ✗ | single inference steps only |
| | Robustness to Abstracted Data | ✗ | clean inputs only |
| | Quantitative / Qualitative | ★ | quantitative only |
| | Open-ended Questions | ✗ | binary + fixed queries |
| LLM-ORBench | Complex Reasoning Chains | ✓ | multiple inference steps |
| | Robustness to Abstracted Data | ✓ | supports abstractions |
| | Quantitative / Qualitative | ✓ | accuracy/conf./halluc. |
| | Open-ended Questions | ✓ | binary + free-form queries |

Table 1: Feature comparison of LLM-ORBench with representative reasoning benchmarks. Symbols: ✓ = full, ★ = partial, ✗ = none.

classification, while ensuring that reasoning remains computationally tractable. This provides a formal and well-defined framework for representing and managing knowledge within the ontology, while clearly defining the scope of expressiveness.

### 3.1.2 MULTI-STEP REASONING

Multi-step reasoning in ontologies refers to the process of deriving new knowledge by applying sequences of logical axioms or triples that extend beyond a single inference step. This capability is essential for both traditional symbolic reasoners and neurosymbolic approaches.

In this work, we advance the complexity of reasoning by *combining multiple inference rules*, thereby enabling deeper multi-step reasoning and generating more comprehensive conclusions. Inference rules are formal logical patterns that allow the derivation of new facts from existing knowledge. For example, given the assertions `X worksAt Y` and the axiom that `worksAt` has range `Organization`, a multi-step inference can proceed as follows: first, from `X worksAt Y` we recognize that `Y` is in the range of `worksAt`; then, applying the range axiom, we infer that `Y` must be an `Organization`.

### 3.2 DATA CREATION

This subsection describes the process of constructing a natural language question-answering dataset using a given ontology to evaluate the reasoning capabilities of LLMs in the context of complex ontology-based reasoning. The structure of our framework is visualized in Figure 1. This subsection will aim to explain how each process was completed to create the benchmark and how LLMs will be evaluated.

### 3.2.1 ONTOLOGY GENERATION

Let $G$ denote the original ontology. For each resource $r$, we construct a subgraph $g$ containing all triples in which $r$ appears as either the subject or object (referred to as *1-hop graphs*). To enable more comprehensive inference, each subgraph $g$ is extended to include all statements within two hops of $r$, yielding a *2-hop graph* $g'$. Two hops were selected as the graph provides sufficient

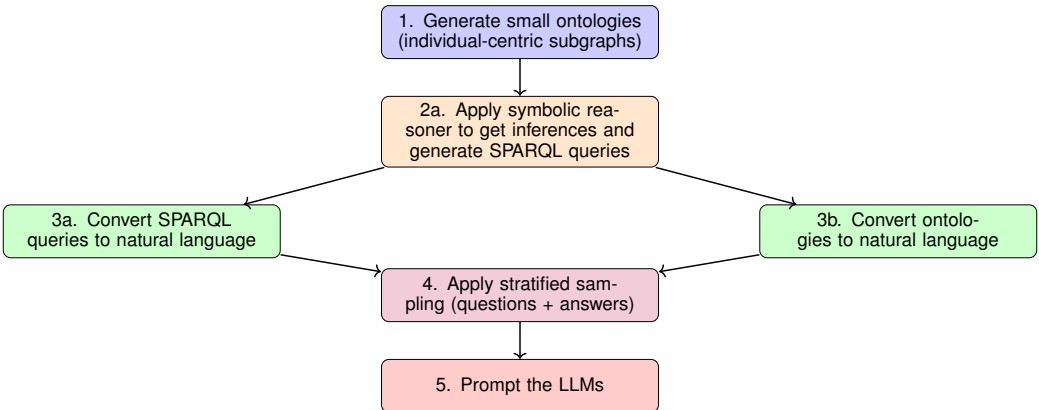

Figure 1: Framework for constructing the benchmark to evaluate LLMs on ontology-based reasoning tasks. Color coding highlights the stages: blue = ontology generation, orange = reasoning, green = natural language conversion, purple = sampling, red = prompting.

richness to enable more complex and meaningful inferences, while remaining compact enough to allow efficient reasoning and facilitate deeper insights. In both 1-hop and 2-hop graphs, the TBox is included to preserve semantic context.

### 3.2.2 QUESTION AND ANSWER GENERATION (FROM SYMBOLIC INFERENCES)

We employ the Pellet reasoner (Sirin et al., 2007) to derive ground-truth inferences from OWL ontologies, encompassing both class membership and property assertions. The reasoner is applied to 1-hop graphs $g$ and 2-hop graphs $g'$, producing the corresponding inference graphs $i$ and $i'$ (hereafter referred to as *1-hop inferences* and *2-hop inferences*, respectively).

The inferences, represented in triple format, are used as benchmark answers. Corresponding SPARQL queries are then generated to serve as benchmark questions and subsequently translated into natural language. For each inference, we define two complementary tasks: (i) Binary Question Answering (BQA) and (ii) Open-Ended Question Answering (OEQA). Figure 2 (see Appendix) provides a detailed overview of the process used to generate ontologies along with their corresponding questions and answers.

### 3.2.3 NATURAL LANGUAGE CONVERSION (VERBALIZATION)

To translate SPARQL queries into natural language, we employed *GPT-4o-mini*, chosen for its cost-effectiveness and efficiency in handling straightforward tasks. A temperature setting of 0 was used to ensure deterministic and consistent outputs. Furthermore, contextual information from the ontologies was also translated into natural language using the same language model.

### 3.2.4 STRATIFIED SAMPLING

To analyze the reasoning patterns underlying each inference, we developed a *comprehensive tagging framework* that categorizes explanations based on the ontological constructs and logical operations involved. The system employs 20 distinct tags to capture different reasoning mechanisms, as summarized in Table 5 (see Appendix).

Explanation paths are encoded as *tag strings*, where repeated reasoning types are represented by repeated tags (i.e., HH for two hierarchy steps, DHT for a direct assertion followed by hierarchy and transitivity). The maximum tag string length for an instance serves as a proxy for reasoning complexity, enabling differentiation between simpler and more elaborate inference chains.

To reduce the number of sampled instances while maintaining diversity in both *size* and *reasoning complexity*, we employed a *stratified sampling* strategy. Instances were grouped into bins along two dimensions: (i) *Complexity:* Measured using the tagging framework, the *maximum tag string*

*length* across all valid explanations for an instance represents the longest sequence of reasoning steps required, and (ii) *Size:* Defined as the number of triples in each 1-hop and 2-hop graph.

Each query was assigned a stratum label combining its complexity and size bins, allowing both dimensions to be captured in a single identifier. Stratified sampling over these labels was then applied to obtain a balanced and representative subset of queries.

## 4 EXPERIMENTS

Our experiments assessed the reasoning capabilities of state-of-the-art large language models (LLMs) using datasets developed within the LLM-ORBench framework. This section outlines the ontologies and LLMs included in our experiments, the different settings used to test various aspects of model performance, and the evaluation metrics employed.

### 4.1 LARGE LANGUAGE MODELS

We evaluate three LLMs from distinct providers—OpenAI, Meta, and DeepSeek—spanning reasoning vs. non-reasoning and open- vs. closed-source paradigms:

- **GPT-5-mini** (snapshot: 2025-08-07): GPT-5-mini (with `reasoning_effort="low"`) serves as a cost-constrained reasoning baseline, chosen to balance accuracy and speed vs. cost.
- **DeepSeek-V3-0324**: A non-reasoning, mixture-of-experts (MoE) model providing performance comparable to LLaMA-4, representing an open-source, alternative architecture.
- **LLaMA-4-Maverick-17B-128E-Instruct**: A non-reasoning MoE model from Meta, closely matched to DeepSeek in evaluation cost and performance, representing another open-source alternative.

### 4.2 ONTOLOGIES

The experiments employ two distinct ontologies, chosen to represent a range of complexities, structural characteristics, and application domains:

- **Family Ontology** (Stevens & Stevens, 2008): A widely used ontology for modeling genealogical knowledge and family relationships. It provides a foundational framework for reasoning over kinship terms, familial roles, and relations such as parent-child, sibling, and spouse. The Family ontology comprises 2,527 axioms.
- **OWL2Bench** (Singh et al., 2020): A benchmark ontology designed to evaluate ontology reasoners in terms of coverage, scalability, and query performance. It extends the University Ontology Benchmark (UOBM) by generating four TBoxes, one for each OWL 2 profile (EL, QL, RL, and DL). OWL2Bench incorporates a diverse set of axioms, including class expressions, object and data property axioms, and assertions, along with an ABox generator and 22 SPARQL queries requiring reasoning tasks. In this work, we use OWL2Bench1-DL, where `1` denotes the number of universities and `DL` the OWL 2 profile. The OWL2Bench1 ontology contains 60,573 axioms.

For each ontology, we generate a set of questions using the LLM-ORBench framework. Descriptive statistics of the sampled datasets are reported in Table 2.

### 4.3 EXPERIMENTAL SETTINGS

To enable a comprehensive and multi-dimensional evaluation, we employ a three-setting prompting strategy. This design allows us to compare LLM reasoning performance across natural language, formal logic, and abstracted representations, revealing how different knowledge and query representations affect reasoning accuracy.

| Ontology | | BQ | OEQ | Sampled BQ | Sampled OEQ | Sampled Questions |
|---|---|---|---|---|---|---|
| Family | 1-hop | 23,043 | 18,256 | 1,159 | 265 | 1,424 |
| | 2-hop | 39,809 | 37,061 | 1,882 | 301 | 2,183 |
| OWL2Bench | 1-hop | 68,826 | 36,681 | 1,969 | 492 | 2,461 |
| | 2-hop | 99,419 | 56,398 | 3,750 | 823 | 4,573 |

Table 2: Descriptive statistics of the sampled datasets used in the experiments. For each ontology, we report the number of binary questions (BQ), open-ended questions (OEQ), and the total number of sampled questions, separately for 1-hop and 2-hop graphs.

- **Natural Language Reasoning**: The model receives a natural language question together with a verbalized ontology. This setting evaluates general natural language understanding and reasoning in a conversational style.
- **Formal-Symbolic Reasoning**: The model is provided with an ontology in formal `OWL` syntax along with a corresponding `SPARQL` query. This setting assesses the model's ability to execute logical reasoning directly against a structured knowledge base.
- **Abstracted Reasoning**: In this setting, the model is provided with an abstracted version of natural language reasoning, where semantic content and domain knowledge are removed to isolate pure logical inference. To achieve this, both the benchmark (questions and answers) and associated contexts (resource-centric subgraphs $g$ and $g'$) are abstracted. Properties and classes are systematically replaced with unique identifiers (i.e., "Property1", "Class2"), and individual instances are renamed sequentially (i.e., "Individual1"), ensuring consistent transformation across SPARQL queries and their contexts before conversion to natural language.

We employ a zero-shot, instruction-only prompt with a fixed output schema. An example of the prompt template is shown in Figure 3 (see Appendix).

## 4.4 EVALUATION METRICS

The reasoning capabilities of LLMs are evaluated using both quantitative and qualitative metrics, providing a comprehensive assessment that extends beyond accuracy alone. To standardize evaluation, models output three components: the final answer, a confidence score in the range $[0, 1]$, and a self-reported estimate of reasoning complexity, expressed as the number of distinct reasoning steps.

Evaluation is conducted using *DeepEval* (Sinha et al., 2025), which implements three core metrics (see Subsection A.2 in Appendix for more details):

- **Jaccard Similarity/Accuracy**: evaluates correctness as the similarity between the sets of expected and predicted answers.
- **Confidence Calibration**: measures the agreement between the model's reported confidence scores and its empirical correctness.
- **Hallucination Detection**: quantifies the tendency of the model to produce answers outside the set of valid options, relevant only for multiple-choice questions.

## 5 RESULTS

This section reports the results of our evaluation of three state-of-the-art LLMs: *GPT-5-mini*, *DeepSeek-V3*, and *LLaMA-4-Maverick*. We examined their reasoning abilities on two benchmark datasets, OWL2Bench and Family, using the LLM-ORBench framework.

Our evaluation examined LLM performance along four key dimensions: (1) reasoning depth (1-hop vs. 2-hop inferences), (2) knowledge representation format (natural language vs. SPARQL), (3) level of abstracted reasoning (semantic vs. abstracted content), and (4) question type (binary vs. open-ended questions). The analysis uncovers substantial performance differences, highlighting both the strengths and the limitations of current LLMs in formal logic.

Tables 3 and 4 summarize the performance across different reasoning settings and question types (i.e., binary and open-ended) on Family and OWL2Bench, respectively. Performance is evaluated using Jaccard accuracy, confidence scores, and, for open-ended questions, hallucination rates.

**Performance on Family ontology:** Table 3 shows that all models perform well on 1-hop binary questions, with Jaccard accuracy ranging from $91.5\%$ (GPT-5-mini) to $95.7\%$ (LLaMA-4-Maverick), but accuracy declines on 2-hop questions ($73.8\%$–$84.9\%$). Formal-symbolic reasoning boosts 2-hop performance for GPT-5-mini ($84.0\%$), while DeepSeek-V3 and LLaMA-4-Maverick show lower gains. Abstracted reasoning is consistently the most challenging, with DeepSeek-V3 achieving only $9.7\%$ on 2-hop questions, and LLaMA-4-Maverick showing overconfidence by attaining $31.1\%$ accuracy with $91.9\%$ confidence.

For open-ended questions, overall accuracy is lower, but patterns are similar: 2-hop inferences reduce performance under NL reasoning, formal-symbolic reasoning sometimes improves results, and abstracted reasoning remains most difficult. Hallucination rates are generally below $20\%$, except for LLaMA-4-Maverick on abstracted tasks ($30.9\%$–$44.0\%$). These findings highlight how reasoning type, question complexity, and abstraction interact to affect both accuracy and reliability.

**(a) Binary Question Performance**

| Model | Setting | Jaccard Accuracy | | Confidence Score | |
|---|---|---|---|---|---|
| | | 1-hop | 2-hop | 1-hop | 2-hop |
| GPT-5-mini | NL Reasoning | 91.5% | 73.8% | 90.9% | 78.5% |
| | Formal-Symbolic Reasoning | 89.0% | 84.0% | 88.9% | 86.1% |
| | Abstracted Reasoning | 25.6% | 23.6% | 61.2% | 53.6% |
| DeepSeek-V3 | NL Reasoning | 93.8% | 84.9% | 93.7% | 83.6% |
| | Formal-Symbolic Reasoning | 90.7% | 80.6% | 90.5% | 79.8% |
| | Abstracted Reasoning | 11.8% | 9.7% | 30.8% | 39.5% |
| LLaMA-4-Maverick | NL Reasoning | 95.7% | 84.5% | 95.3% | 87.4% |
| | Formal-Symbolic Reasoning | 89.4% | 73.8% | 88.0% | 73.1% |
| | Abstracted Reasoning | 39.4% | 31.1% | 89.7% | 91.9% |

**(b) Open-Ended Question Performance**

| Model | Setting | Jaccard Accuracy | | Confidence Score | | Hallucination Rate | |
|---|---|---|---|---|---|---|---|
| | | 1-hop | 2-hop | 1-hop | 2-hop | 1-hop | 2-hop |
| GPT-5-mini | NL Reasoning | 70.4% | 55.1% | 41.4% | 32.1% | 3.5% | 6.4% |
| | Formal-Symbolic Reasoning | 80.0% | 62.3% | 57.6% | 43.1% | 2.1% | 3.5% |
| | Abstracted Reasoning | 47.2% | 26.5% | 46.9% | 43.8% | 8.8% | 15.3% |
| DeepSeek-V3 | NL Reasoning | 75.3% | 54.5% | 41.3% | 26.6% | 5.1% | 6.5% |
| | Formal-Symbolic Reasoning | 57.4% | 55.4% | 42.1% | 36.1% | 11.8% | 11.2% |
| | Abstracted Reasoning | 39.1% | 42.1% | 17.2% | 16.0% | 18.8% | 14.1% |
| LLaMA-4-Maverick | NL Reasoning | 63.8% | 44.2% | 39.3% | 22.9% | 8.8% | 8.3% |
| | Formal-Symbolic Reasoning | 68.4% | 55.3% | 41.4% | 29.8% | 11.1% | 12.6% |
| | Abstracted Reasoning | 29.1% | 13.4% | 41.5% | 45.6% | 30.9% | 44.0% |

Table 3: Performance of LLMs on Family ontology.

**Performance on OWL2Bench ontology** Similar trends are observed as with Family, but accuracy is notably lower on OWL2Bench. Table 4 shows that all models achieve moderate performance on 1-hop binary questions, with Jaccard accuracy ranging from $44.6\%$ (DeepSeek-V3) to $66.9\%$ (LLaMA-4-Maverick), and declines further on 2-hop questions ($37.0\%$–$50.3\%$). Formal-symbolic reasoning underperforms, with GPT-5-mini reaching only $19.9\%$ on 2-hop questions. Abstracted reasoning is the most challenging, with accuracy between $4.6\%$ and $36.2\%$, although LLaMA-4-Maverick maintains high confidence ($72.8\%$–$90.1\%$), indicating overconfidence.

For open-ended questions, accuracy is substantially lower overall, yet trends remain consistent. Hallucination rates are high in several cases, particularly for DeepSeek-V3 and LLaMA-4-Maverick, reaching up to $73.7\%$ under NL reasoning. Confidence scores do not consistently align with accuracy, with some models displaying high confidence even for low-accuracy tasks. These results

underscore the difficulty of OWL2Bench for multi-step and abstract reasoning, highlighting both accuracy limitations and prevalent overconfidence and hallucinations.

**(a) Binary Question Performance**

| Model | Setting | Jaccard Accuracy | | Confidence Score | |
|---|---|---|---|---|---|
| | | 1-hop | 2-hop | 1-hop | 2-hop |
| GPT-5-mini | NL Reasoning | 60.4% | 47.8% | 66.1% | 62.8% |
| | Formal-Symbolic Reasoning | 32.2% | 19.9% | 39.5% | 28.9% |
| | Abstracted Reasoning | 18.6% | 12.7% | 49.3% | 66.2% |
| DeepSeek-V3 | NL Reasoning | 44.6% | 37.0% | 59.6% | 50.6% |
| | Formal-Symbolic Reasoning | 37.1% | 28.1% | 30.6% | 28.1% |
| | Abstracted Reasoning | 8.6% | 4.6% | 26.3% | 51.5% |
| LLaMA-4-Maverick | NL Reasoning | 66.9% | 50.3% | 80.1% | 83.2% |
| | Formal-Symbolic Reasoning | 30.1% | 21.6% | 24.1% | 21.3% |
| | Abstracted Reasoning | 36.2% | 20.3% | 72.8% | 90.1% |

**(b) Open-ended Question Performance**

| Model | Setting | Jaccard Accuracy | | Confidence Score | | Hallucination Rate | |
|---|---|---|---|---|---|---|---|
| | | 1-hop | 2-hop | 1-hop | 2-hop | 1-hop | 2-hop |
| GPT-5-mini | NL Reasoning | 28.1% | 22.0% | 58.7% | 68.9% | 46.2% | 44.3% |
| | Formal-Symbolic Reasoning | 27.1% | 29.2% | 8.3% | 62.5% | 3.4% | 21.4% |
| | Abstracted Reasoning | 3.1% | 7.2% | 69.3% | 74.5% | 44.1% | 47.5% |
| DeepSeek-V3 | NL Reasoning | 35.3% | 32.2% | 11.5% | 8.2% | 13.8% | 6.1% |
| | Formal-Symbolic Reasoning | 30.3% | 32.6% | 13.9% | 8.1% | 50.6% | 47.3% |
| | Abstracted Reasoning | 22.4% | 28.1% | 6.7% | 17.0% | 65.5% | 61.9% |
| LLaMA-4-Maverick | NL Reasoning | 17.9% | 8.5% | 28.9% | 39.9% | 56.4% | 73.7% |
| | Formal-Symbolic Reasoning | 4.9% | 2.7% | 8.1% | 7.9% | 61.4% | 67.2% |
| | Abstracted Reasoning | 1.1% | 0.8% | 56.4% | 44.6% | 59.1% | 61.5% |

Table 4: Performance of LLMs on OWL2Bench ontology.

## 6 LIMITATIONS

Despite providing valuable insights, this study has several limitations. The benchmarks used may not capture the full scale and diversity of real-world ontologies. The performance drop on OWL2Bench highlights the challenges posed by larger, more complex structures. Furthermore, the current framework allows for "shortcut reasoning", where models can arrive at correct answers without performing genuine multi-step inference, potentially inflating results. These limitations highlight the need for a more diverse, challenging, and rigorously designed benchmark to fully assess reasoning capabilities.

To encourage more sophisticated reasoning, we plan to modify the benchmark by selectively removing information that allows for trivial or direct conclusions. Eliminating these "shortcuts" should compel models to perform deeper, multi-step inference rather than relying on straightforward paths. Another area for improvement is the introduction of controlled noise, such as misleading or partially incorrect information, to test model robustness. This approach would allow us to better evaluate reasoning under uncertainty and assess the models' ability to navigate ambiguous or conflicting data.

## 7 CONCLUSION

In summary, *LLM-ORBench* provides a structured and systematic approach to evaluate the formal reasoning capabilities of LLMs in ontology-based tasks. Our experiments reveal that while current LLMs can handle certain multi-step reasoning scenarios, their performance declines significantly under abstraction and in open-ended question-answering tasks, often exhibiting overconfidence or hallucinations. Overall, this work underscores the importance of specialized benchmark frameworks like LLM-ORBench for uncovering the strengths and weaknesses of LLMs in formal logic.

## REPRODUCIBILITY STATEMENT

The full implementation and datasets are available at `https://anonymous.4open.science/r/LLM-ORBench-8F37/README.md`.

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

# A SUPPORTING MATERIALS

## A.1 FIGURES AND TABLES

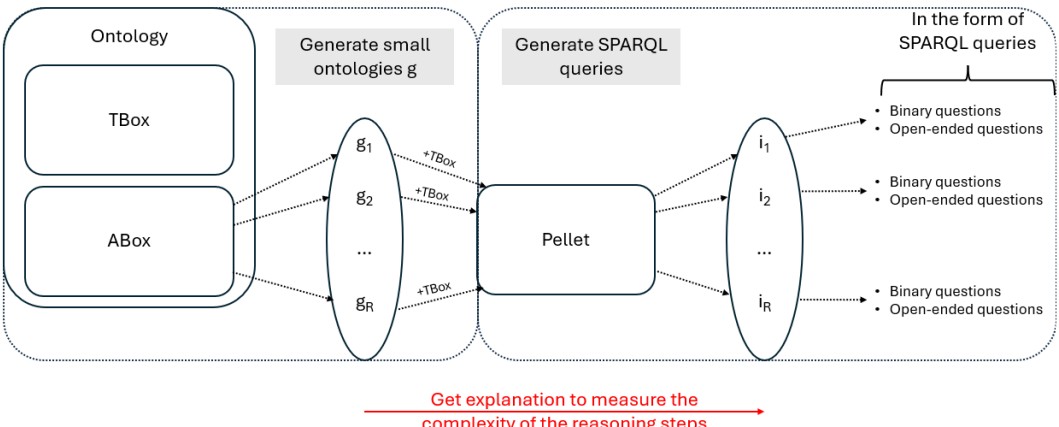

Figure 2: Detailed process for generating ontologies and corresponding SPARQL/NL queries.

| | | | |
|---|---|---|---|
| **D** | Direct assertions | **I** | Inverse properties |
| **H** | Hierarchy | **F** | Functional properties |
| **T** | Transitivity | **V** | Reflexive properties |
| **S** | Symmetry | **Y** | Irreflexive properties |
| **A** | Asymmetric properties | **Q** | Equivalence |
| **J** | Disjointness constraints | **R** | Domain / range restrictions |
| **N** | Property chains | **C** | Cardinality restrictions |
| **E** | Existential restrictions | **L** | Universal restrictions |
| **∩** | Intersection | **∪** | Union |
| **M** | Multi-step reasoning | | |

Table 5: Abbreviations of the tags employed in our comprehensive tagging framework.

```
[Instruction Header: Tailored to the experimental setting.]

ANSWER: [The model's final answer]      % TRUE/FALSE or open-ended
CONFIDENCE: [Score between 0.0 and 1.0 indicating certainty]
REASONING_STEPS: [Number of distinct reasoning steps applied]

Question: {QUESTION}                     % Natural language or SPARQL
Context: {CONTEXT}                        % Verbalized or OWL ontology
```

Figure 3: An example of the prompt template.

## A.2 EVALUATION METRICS

### A.2.1 JACCARD SIMILARITY COEFFICIENT (JACCARD ACCURACY)

We employ the Jaccard similarity coefficient (Tan et al., 2005; Manning et al., 2008) to compare expected and predicted answers. For binary questions (TRUE/FALSE), this reduces to exact match scoring (0 or 1). For open-ended questions, it measures the intersection over union of answer sets, thereby granting partial credit when a model identifies only a subset of the correct responses.

The Jaccard similarity coefficient is defined as:

$$J(A, E) = \frac{|A \cap E|}{|A \cup E|} \tag{1}$$

where $A$ denotes the set of predicted answers and $E$ the set of expected answers.

### A.2.2 CONFIDENCE CALIBRATION SCORE (CONFIDENCE SCORE)

We also applied a confidence calibration score, which evaluates a model's ability to estimate its own uncertainty by comparing self-reported confidence scores (ranging from 0.0 to 1.0) with actual performance. A well-calibrated model's confidence scores should accurately reflect its likelihood of being correct. The calibration score is computed as:

$$\text{Calibration Score} = 1 - |C - J| \tag{2}$$

where $C$ denotes the model's confidence score and $J$ the Jaccard similarity coefficient. A calibration score above $0.7$ indicates good calibration.

This approach aligns with established calibration metrics in machine learning, such as the Expected Calibration Error (ECE) (Guo et al., 2017; Wang et al., 2024), which measures the discrepancy between predicted probabilities and actual outcomes.

### A.2.3 HALLUCINATION RATE

Finally, we also calculated an hallucination rate, which quantifies the tendency of models to generate answers that are not valid options for a given query. This metric is applied exclusively to open-ended questions, where a hallucination is defined as any entity or value absent from the set of correct answer options.

For open-ended questions, the hallucination score is calculated as:

$$H = \frac{\text{Number of Hallucinated Answers}}{\text{Total Answers Provided}} \tag{3}$$

where a higher score indicates a greater propensity for producing invalid or unsupported answers.

For binary questions (TRUE/FALSE), hallucination is not applicable, since the model can only select from two predefined options.

## B THE USE OF LARGE LANGUAGE MODELS (LLMS)

We have used ChatGPT to address the grammatical errors and rephrase the sentences.

