# OpenReview forum: "LLM-ORBench: Designing a Benchmark Dataset for Complex Ontology-Based Reasoning Tasks in Large Language Models"
_ICLR.cc/2026/Conference — ICLR 2026 Conference Withdrawn Submission_

### Official Review · Reviewer_2BVf · 2025-10-31

**Soundness:** 3
**Presentation:** 2
**Contribution:** 2
**Rating:** 4
**Confidence:** 2

**Summary:**

This paper propose a benchmark to evaluate LLM's logical reasoning capability in isolation, Specifically, this work generated questions from ontologies, e.g. family relationships. First, a symbolic reasoner is used to generate SPQRQL queries, and then the queries are translated into natural language format using LLMs.

Within this framework, the authors can control several perspective of the generated queries, including the complexity.

**Strengths:**

This work targets the pure logical reasoning capability of LLMs, which is an important perspective of the modern LLM research.

The authors propose a viable approach to generate logical reasoning queries based on symbolic reasoner.

**Weaknesses:**

1. Only a small number of models are evaluated. I think at least more open-source models should be evaluated. Otherwise, the results may not be representative enough.

2. The generated dataset, although isolating logical reasoning parts from the domain knowledge, may not reflect real-world scenarios. the scenarios are pretty restricted. So it requires justification on why this proposed benchmark is practically relevant.

3. This work resembles some existing synthetic benchmarks using tree or graph for generating logical reasoning queries, so the related work could be enriched.

4. The dataset generation process is not clear enough. It would be better to give a concrete example of the generation process, including what does the ontology look like, how is this used to generate symbolic queries, so on and so forth.

**Questions:**

Please explain more on the stratified sampling. It is understandable to tag the explanations for understanding the inference process. How are these tags obtained? By LLMs? What is the sampling used for?

---

### Official Review · Reviewer_dS4s · 2025-10-31

**Soundness:** 3
**Presentation:** 3
**Contribution:** 3
**Rating:** 6
**Confidence:** 3

**Summary:**

This work introduces a novel benchmark called LLM-ORBench for evaluating LLMs' reasoning capabilities on ontologies. Compared to the existing benchmarks, the new benchmark includes open-ended questions and supports evaluation beyond quantitative metrics.

**Strengths:**

1. A new benchmark for ontology-based reasoning evaluation can provide useful resources to the community. It also uses real-world ontologies (Family and OWL2Bench) rather than purely synthetic ones.
2. The inclusion of qualitative metrics and open-ended questions seems novel compared to existing benchmarks.
3. Systematic abstraction of domain knowledge is a novel approach to test pure logical reasoning. Experiment and analysis demonstrate a gap in isolating LLMs' inference capabilities from pre-trained world knowledge.

**Weaknesses:**

1. Degradation in the abstracted setting (Section 4.3)  performance may be due to unnatural surface forms (e.g., “Property17,” “Individual3”), tokenization artifacts, and degraded linguistic priors. The observed performance drop may stem from distribution shift and unnaturalness, not purely from isolation of formal reasoning. No control experiments test this. For instance, since Family and OWL2Bench are publicly available, models may have memorized their structures or example queries. Replacing ontology terms with meaningless tokens like “foo” “bar” could help validate this.
2. Hallucination metric is inconsistently defined (Section 4.4 says it’s “relevant only for multiple-choice,” yet applied to open-ended questions), and it depends on a “valid options” set that is not described for free-form NL.
3. No analysis of input length and context size effects. OWL2Bench drops may be due to longer contexts rather than deeper reasoning; no ablations control for context length.
4. Claimed contribution “comparison with a symbolic reasoner” (Contribution 3) is not covered in experiments.
5. The benchmark tops out at 2-hop subgraphs (Section 3.2.1), yet claims “complex, multi-step reasoning.” The contribution may be overstated.

**Questions:**

1. The paper evaluates with "reasoning_effort='low'" for GPT-5-mini (Section 4.1). Were higher effort levels tested, and how would this affect the abstracted reasoning performance?
2. Are the generated SPARQL queries guaranteed to be correct?
3. Writing: inconsistent capitalization, “PRONTOQA” vs “ProntoQA”
4. See above "Weaknesses" for other concerns

---

### Official Review · Reviewer_Ha8p · 2025-11-01

**Soundness:** 2
**Presentation:** 2
**Contribution:** 2
**Rating:** 2
**Confidence:** 3

**Summary:**

This paper introduces LLM-ORBench, a new benchmark for systematically evaluating large language models (LLMs) on complex ontology-based reasoning tasks. An ontology is defined as a formal and explicit specification of shared concepts within a domain. The benchmark assesses LLMs’ ability to perform multi-step logical and abstract reasoning without relying on domain knowledge. The authors highlight current issues of hallucination, lack of domain-specific knowledge, and poor interpretability in LLM reasoning. Compared with previous benchmarks, LLM-ORBench improves in four key aspects: complex reasoning chains, robustness to abstracted data, combined quantitative and qualitative metrics, and support for open-ended questions. Using the Family Ontology and OWL2Bench, the authors construct a large-scale benchmark dataset and evaluate three models: GPT-5-mini, DeepSeek-V3-0324, and LLaMA-4-Maverick-17B-128E-Instruct.

**Strengths:**

1.	The benchmark is large-scale, encompassing a wide range of ontology-based reasoning questions.

2.	The study evaluates three LLMs of varying architectures and sizes to comprehensively test the benchmark.

3.	LLM-ORBench advances prior benchmarks by enhancing Complex Reasoning Chains and Robustness to Abstracted Data, enabling more accurate assessment of LLM reasoning capabilities.

4.	The authors conduct comparative experiments across natural language, formal logic, and abstracted representations to analyze how input formats affect reasoning accuracy and robustness.

5.	A systematic data construction pipeline—using 2-hop subgraph generation, the Pellet reasoner, and stratified sampling—ensures reliable ground-truth inferences and controlled task difficulty.

**Weaknesses:**

1. I find the motivation of this paper somewhat unclear. The authors claim that the proposed benchmark aims to address common issues in LLM reasoning—namely hallucinations, lack of domain-specific knowledge, and poor interpretability. However, the paper does not clearly explain why existing benchmarks fail to adequately address these issues, nor does it provide a detailed analysis of how LLM-ORBench specifically mitigates them. A more explicit discussion contrasting prior benchmarks’ limitations with the proposed design choices would substantially strengthen the motivation and clarify the unique contribution of this work.

2. I am fairly certain that there exists a large body of work and numerous benchmarks in the area of formal reasoning. However, the paper only references and discusses a few of them. Could the authors provide a more comprehensive comparison with other relevant benchmarks and works?[1,2,3,4,5] A deeper analysis would help clarify what specific pain points or limitations exist in current ontology-based reasoning problems, and how LLM-ORBench addresses them. I believe that more thorough analysis and discussion are necessary.

3. The authors claim to provide “an empirical comparison of multiple LLMs and a well-known symbolic reasoner across diverse ontology reasoning tasks and benchmark variations.” However, the experiments only include three LLMs, GPT-5-mini, DeepSeek-V3-0324, and LLaMA-4-Maverick-17B-128E-Instruct. Could the authors provide additional results on more LLMs (different scales) to further validate the usefulness and generality of the proposed benchmark?

4. According to Table 3, although model performance declines on 2-hop questions, the Jaccard accuracy and confidence scores remain relatively high for most binary questions. This raises the concern of whether the proposed benchmark truly possesses sufficient discriminative power to differentiate the reasoning capabilities of different models.

5. In line 246, the paper mentions translation, but it is not immediately clear whether this refers to a traditional rule-based or a large language model (LLM)-based approach. Section 3.2.3 later states that GPT-4o-mini was used for translation from formal (SPARQL/OWL) to natural language. However, relying on an LLM for this step could substantially reduce the reliability and controllability of LLM-ORBench. Have there been prior works showing that current LLM-based translation methods are sufficiently reliable for formal-to-natural conversions? I believe more justification and discussion on this aspect are necessary.

6. This benchmark highlights open-ended questions as one of its main advantages compared to previous benchmarks. However, in Section 4.4, the metrics reported for evaluating accuracy or correctness are limited to Jaccard similarity and accuracy. It remains unclear how the results of open-ended questions are specifically evaluated under this framework. In my view, open-ended settings pose inherent challenges for evaluating reasoning ability, as it is often difficult to objectively assess correctness or reasoning quality when responses are free-form and potentially diverse in expression and structure.

[1] Oyvind Tafjord, Bhavana Dalvi, and Peter Clark. 2021. ProofWriter: Generating implications, proofs, and abductive statements over natural language. In Findings of the Association for Computational Linguis tics: ACL-IJCNLP 2021, pages 3621–3634, Online. Association for Computational Linguistics.

[2] Mohammed Saeed, Naser Ahmadi, Preslav Nakov, and Paolo Papotti. 2021. RuleBERT: Teaching soft rules to pre-trained language models. In Proceedings of  the 2021 Conference on Empirical Methods in Natural Language Processing, pages 1460–1476, Online and Punta Cana, Dominican Republic. Association for Computational Linguistics.

[3] Pratik Joshi, Somak Aditya, Aalok Sathe, and Monojit Choudhury. 2020. TaxiNLI: Taking a ride up the NLUhill. In Proceedings of the 24th Conference on  Computational Natural Language Learning, pages 41–55, Online. Association for Computational Linguistics.

[4] Simeng Han, Hailey Schoelkopf, Yilun Zhao, Zhenting Qi, Martin Riddell, Luke Benson, Lucy Sun, Ekate rina Zubova, Yujie Qiao, Matthew Burtell, et al. 2022.  FOLIO: Natural language reasoning with first-order logic. arXiv preprint arXiv:2209.00840.

[5] Mohammed Saeed, Naser Ahmadi, Preslav Nakov, and Paolo Papotti. 2021. RuleBERT: Teaching soft rules to pre-trained language models. In Proceedings of  the 2021 Conference on Empirical Methods in Natural Language Processing, pages 1460–1476, Online and Punta Cana, Dominican Republic. Association for Computational Linguistics.

**Questions:**

1. This work designs the benchmark based solely on two ontologies—Family Ontology and OWL2Bench. Can these two ontologies be considered representative enough to generalize across broader ontology-based reasoning tasks? The authors should provide more justification or discussion on this point. In my opinion, the benchmark should aim to evaluate ontology-based reasoning problems rather than ontology-specific reasoning problems, and thus its generality and representativeness are crucial to its validity.

2. Compared to previous benchmarks that focus mainly on quantitative evaluation, the authors emphasize that LLM-ORBench additionally incorporates qualitative evaluation as one of its advantages. My question, however, is whether reasoning ability should primarily be assessed through the accuracy or correctness of generated answers rather than their linguistic quality. In other words, can a response that is more fluent or rhetorically polished truly indicate stronger reasoning ability? Furthermore, the three metrics described in Section 4.4—Jaccard Accuracy, Confidence Calibration, and Hallucination Rate—all seem to be quantitative measures.

3. In the experiments, would incorporating a comparison with a symbolic reasoner provide stronger reference points for assessing the reliability and discriminative validity of LLM-ORBench?

---

### Note · Authors · 2026-02-24

I have read and agree with the venue's withdrawal policy on behalf of myself and my co-authors.

---

### Meta-Review · Area_Chair_vTBr · 2026-01-07

**Summary:**

This paper introduces LLM-ORBench, a benchmark for evaluating ontology-based reasoning in LLMs with large-scale, systematic data construction using real-world ontologies and a symbolic reasoner. It introduces a novel evaluation dimension to isolate logical reasoning from world knowledge.

However, the reviewers raise significant concerns that must be addressed for the paper to be accepted:
1. The justification for the new benchmark is unclear. The authors must provide a more thorough and critical analysis of related work (especially existing reasoning benchmarks) to sharply delineate the specific gaps LLM-ORBench fills and clarify its relevance of real world scenarios.
2. It is essential to include more LLMs for evaluation to demonstrate the benchmark's discriminative power and generalization. Besides，the metrics reported for evaluating are limited for open question and the definition is not consistent.
3. The dataset generation process is not clear enough and ambiguous without concrete examples.

**Reviewer Concerns:**

The authors did not submit any rebuttal.

**Reviewer Scores:**

None has changed score.

---

### Decision · Program_Chairs · 2026-01-26

Reject